# Accurate 3D to 2D Object Distance Estimation from the Mapped Point Cloud Data

**DOI:** 10.3390/s23042103

**Published:** 2023-02-13

**Authors:** Saidrasul Usmankhujaev, Shokhrukh Baydadaev, Jang Woo Kwon

**Affiliations:** Department of Electronic and Computer Engineering, Inha University, Incheon 22212, Republic of Korea

**Keywords:** 3D object detection, deep neural networks, sensor fusion, computer vision

## Abstract

Distance estimation is one of the oldest and most challenging tasks in computer vision using only a monocular camera. This can be challenging owing to the presence of occlusions, noise, and variations in the lighting, texture, and shape of objects. Additionally, the motion of the camera and objects in the scene can affect the accuracy of the distance estimation. Various techniques have been proposed to overcome these challenges, including stereo matching, structured light, depth from focus, depth from defocus, depth from motion, and time of flight. The addition of information from a high-resolution 3D view of the surroundings simplifies the distance calculation. This paper describes a novel distance estimation method that operates with converted point cloud data. The proposed method is a reliable map-based bird’s eye view (BEV) that calculates the distance to the detected objects. Using the help of the Euler-region proposal network (E-RPN) model, a LiDAR-to-image-based method for metric distance estimation with 3D bounding box projections onto the image was proposed. We demonstrate that despite the general difficulty of the BEV representation in understanding features related to the height coordinate, it is possible to extract all parameters characterizing the bounding boxes of the objects, including their height and elevation. Finally, we applied the triangulation method to calculate the accurate distance to the objects and statistically proved that our methodology is one of the best in terms of accuracy and robustness.

## 1. Introduction

Owing to the significant advancements in automotive LiDAR sensors in recent years, the processing of point cloud data has become increasingly crucial for autonomous driving. LiDAR sensors can transmit 3D points of the immediate surroundings in real time and excel in extracting distance information, whereas the camera outperforms LiDAR in capturing richer and denser perceptions [1]. Distance estimation is an important problem in the context of autonomous driving. The challenge of 3D to 2D object distance estimation is that it involves inferring the depth information of an object from a 2D image, which can be difficult because of the lack of explicit depth cues in the image. This problem is known as the “depth ambiguity” problem and can be caused by factors such as occlusion, texture, and lighting [2]. Additionally, estimating the distance of an object in an image can be affected by the intrinsic and extrinsic parameters of the camera, such as the focal length, sensor size, and lens distortion. A distance estimation system must satisfy several criteria, including a low error magnitude and high reaction rate, to be considered suitable for driving applications. The designed distance estimation system cannot guarantee passenger safety if these requirements are not met [3].

Deep neural networks can easily recognize features obtained from vision sensors. Stereo vision systems can precisely calculate the vast distances between objects. However, they require a long execution time and include a significant amount of computing complexity owing to the calibration and matching between the two cameras, thereby demonstrating low efficiency. Compared with stereo vision, monocular vision can handle complex algorithms and produce the best results faster [4]. In addition, a monocular camera has limitations in terms of the distance estimation of the object; however, sparse point cloud data, obtained from LiDAR, can provide the required 3D information to project monocular perception. Therefore, in this study, we combined the visual perception of a single monocular camera with LiDAR to estimate the distance to a certain object.

Using LiDAR point clouds has various advantages because they are dispersed throughout the measurement area. In addition, point clouds are sparse with varied densities, and the processing of point clouds cannot be affected by basic transformations [5]. LiDAR sensors can be used to calculate distance in three different ways:Triangulation measurement systems consist of a camera and laser transmitter positioned at a fixed angle. Both the position and distance of the laser transmitter from the camera are known. On the target item, the laser emits a pattern that is visible in the camera image. The point appears at a different spot in the camera’s frame of vision depending on the distance from the surface. The distance between the laser source and target object can then be calculated using trigonometry.Time-of-flight measurement systems are related to the time required by a laser pulse to travel from the moment it is emitted to the time the reflected light of the target is detected. The distance traveled by light can then be calculated using the speed of light and refractive index of the medium. They are employed in long-range distance estimates for applications, such as space LiDAR.The phase-shift measurement system demands the use of continuous lasers, which can be modulated to determine the phase difference between the two beams.

To estimate the distance of an object from the camera, the 3D object detection model must first detect the object in the scene. After detecting an object, the distance can be estimated using techniques such as triangulation, which involves using the known dimensions of the object and its apparent size in the image to calculate the distance. Other techniques such as monocular distance estimation are highly based on accurate depth estimation. In addition, stereo vision can be used to estimate distances using 3D object detection. Another methodology, called triangulation, is used in navigation to determine the location of a point by measuring angles to it from known points at each end of a fixed baseline, rather than measuring distances to the point directly. This allows the location of the point to be determined even if it is not possible to measure distances to it directly, for example, if it is located beyond the range of a measuring device or if it is occluded by an obstacle. Subsequently, the location of this point can be determined using trigonometric calculations. After determining the location of the detected object, its distance can be calculated using the Pythagorean theorem. In summary, distance estimation from 3D object detection is a complex task that involves combining computer vision and deep learning techniques to accurately estimate the distance between objects in a scene.

For this study, we used the KITTI dataset [6], which is a pioneering project that suggests a common paradigm for data collection and annotation, such as fitting a car with cameras and LiDAR sensors, using the vehicle to collect data while driving on roads, and using the data to annotate 3D objects [7,8,9]. We mapped sparse point cloud data into bird’s eye view (BEV) images as deep learning frameworks that can be effectively used for feature extraction and inference [10]. When the object was detected by the simplified YOLOv4 detector, we fit the bounding box onto the BEV map and determined the distance to the object using triangulation distance measurement [11]. Subsequently, we performed the Euler region proposal described by Simon et al. [1] to project 3D bounding boxes onto the 2D image.

The main contributions of this paper are as follows:We present robust point cloud processing for distance estimation using a BEV map and image-based triangulation;We propose an image-based approach for metric distance estimation from mapped point cloud data with 3D bounding box projection to the image using the E-RPN model;We demonstrate the viability of extracting all parameters defining the bounding boxes of the objects from the BEV representation, including their height and elevation, even though the BEV representation generally struggles to grasp the features related to the height coordinate.

## 2. Related Work

Over the past decade, numerous studies have been conducted on the perception system as a crucial part of contemporary autonomous driving pipelines, because it aims to precisely evaluate the state of the surrounding environments and provides correct observations for planning and prediction [9,12]. Therefore, 3D object detection for autonomous driving is the main topic; however, few studies have conducted distance estimation for detected objects from the KITTI dataset. In this section, we describe the well-established detection and distance estimation methodologies.

### 2.1. 3D Object Detection

3D object detection methodologies can be classified into three major categories, as shown in Figure 1 [12]:
LiDAR-based detection;
Data representations (points, grids, point voxels, and ranges);Using learning objectives.Camera-based detections (monocular 3D object detection);Multimodal detections.

#### 2.1.1. Point-Based Detection

LiDAR-based detection requires the extraction of features from the sparse and erratically distributed 3D representation of point cloud data, and specific models must be created. Therefore, directly using traditional convolutional networks may not be the best solution. A good example of point-based detection is the analysis of geometric point sets. Qi et al. proposed PointNet++ [13,14], which is an extension of the original PointNet model that uses a hierarchical neural network to handle large-scale point clouds and provides robustness and detailed capture by using neighborhoods at different scales. Using random input dropout during training, the PointNet++ network learns to incorporate multiscale characteristics and weight patterns found at various scales based on the input data. Another example of a sparse-focused detector is StarNet [15] in which the model randomly retrieves a slight subset of neighboring points and classifies the area, features the point cloud, and regresses bounding box parameters. According to the chosen location, the location of the object is projected using only the local data. Because of this configuration, the detector can process each spatial position independently. However, the following two factors primarily limit the representation of point-based detectors: the number of context points and context radius used in feature learning. Richer representation points drastically increase the detection accuracy, although at the expense of significantly more memory usage.

#### 2.1.2. Grid-Based 3D Detection

Grid-based 3D detection refers to a grid-like structure that can be used to identify objects in a scene based on their spatial locations relative to the grid. In this approach, a 3D space is divided into a series of equally sized cubes or cells, and each cell is checked for the presence of an object. If an object is detected, in a particular cell, it is labeled and its location within the grid is recorded. Grid-based neural networks can also be applied to BEV representations [1,8,10,16]. The projected point cloud data construct a grid map from a bird’s eye perspective.

Figure 2a illustrates the sparse depth map that is projected onto the image using a corresponding LiDAR range value; Figure 2b–d illustrate the 360 degree panoramic views for depth, height, and reflectance extracted from the LiDAR data, respectively; and Figure 2e illustrates the BEV map. Chen et al. applied multiview LiDAR to image conversion for their MV3D detector [17]. They used LiDAR to BEV view, LiDAR panoramic (front) images, and RGB images as inputs for their network. This sparsity depends on the number of beams that are mapped to a pixel. The points in three-dimensional space must be projected onto an unwrappable cylindrical surface to flatten the panoramic view of a LiDAR sensor into a two-dimensional image.

### 2.2. Distance Estimation from 3D Object Detection

The majority of distance estimation techniques are based on the “camera-only” technique in which authors rely significantly on monocular distance estimation by applying combined deep neural networks to monocular images. For example, Lee et al. [18] proposed a framework consisting of a detector, depth estimator, and distance predictor. The mean depth, minimum depth, and maximum depth of the object are then extracted by overlapping the bounding box with the depth map. Based on the type of item, bounding box, including its coordinates and size, and extracted depth information, the distance predictor estimates the distance between the object and the camera. Davydov et al. [3] presented a compact convolutional deep learning model that can extract object-specific distance data from monocular images. Their method relies on the models proposed in [7,8]. The pose deviations between the pairs of input images are extracted using models that combine a residual feature extractor and pose estimation network. These networks generate features that are processed using pixel-wise minimum reprojection loss in a self-supervised manner to extract the disparity information. The details of the device configurations utilized for data gathering can be used to further transform disparity maps that have been created into depth maps. They proposed a convolutional neural network model that can extract distance information from the dimensions and appearance of road agents with an optics-based approach of translating the proposed network’s outputs to metric distance values that consider bounding box uncertainty. However, the major limitation of their proposed method is that the mean average error (MAE) [1] for the distance estimation is 6% higher on average, and the model’s error values do not drop below the 2 m point. Huang et al. [4] suggested monocular vision-based distance estimation by identifying and segmenting the target vehicle using an angle regression model (ARN) to extract the target vehicle’s attitude angle information. Their distance estimation is based on the angle information from vehicle recognition and segmentation techniques. It creates the “area-distance” geometric model based on the camera projection concept to calculate the distance to the car in front of it. According to their experiments, their strategy can be used in most traffic situations and shows good robustness against various driving conditions of vehicles in front. However, they are limited in that they solely focused on detecting vehicles and utilized their own set of “rules” to categorize the KITTI dataset.

Another method for distance estimation is based on sensor-fusion technology. According to Kumar et al. [19], an accurate sensor fusion strategy can be used to calculate the distance between a self-driving vehicle and any other car, object, or signboard on its route. A 3D marker was used to accomplish low-level sensor fusion between a camera and LiDAR system based on geometrical transformation and projection. They suggested a method for detecting nearby items by fusing data from a camera and LiDAR, as well as a method for estimating the distance to objects by fusing data on a real road. Additionally, the authors assessed the effectiveness of the data fusion strategy in a few experiments using an autonomous vehicle platform.

## 3. Proposed Method

### 3.1. KITTI Dataset

The KITTI dataset was selected because it is one of the most popular benchmarks among researchers studying monocular and stereo depth estimation, 3D detection, BEV detection, and optical flow evaluation, because it contains superior ground truth data than other datasets. The KITTI dataset includes point cloud data collected from the Velodyne HDL64, which is a high-resolution and high-performance LiDAR sensor that utilizes 64 LiDAR channels with a vertical field of view of 26.9° and delivers a real-time 360° horizontal field of view. The rotation speed ranges from 5 to 20 Hz, enabling the user to assess the density of the data points produced by the LiDAR sensor. It has a range of up to 120 m and can produce point clouds of up to 2,200,000 points per second.

### 3.2. Point Cloud Conversion

The extrinsic and intrinsic parameters were obtained based on the placement of the mounted sensors. Then, the projection matrix, M, is calculated as follows:(1)$$M=M_{I}\;\left(\begin{array}{cccc} f & 0 & u_{0} & 0\\ 0 & f & v_{0} & 0\\ 0 & 0 & 0 & 1 \end{array}\right)\times \;M_{E}\;\left(\begin{array}{cccc} r_{11} & r_{12} & r_{13} & t_{x}\\ r_{21} & r_{22} & r_{23} & t_{y}\\ r_{31} & r_{32} & r_{33} & t_{z}\\ 0 & 0 & 0 & 1 \end{array}\right)$$
where $M_{I}$ and $M_{E}$ are the matrix representations of the intrinsic and extrinsic values, respectively. After estimating the intrinsic and extrinsic parameters, the LiDAR data points were projected onto the camera image. The extrinsic parameters were defined by the six degrees of freedom (6DOF) relative transformation between the LiDAR and camera for data fusion. The gradient of points on the sparse RGB image represents the distance difference from LiDAR to the object (red points represent closer objects and blue points farther objects) (Figure 2). To project a point from the point cloud coordinates onto the color image, the following matrix multiplication is used:(2)$$x=P_{i}\left(\begin{array}{cccc} f_{i} & 0 & u_{i} & -f_{i}b_{x}\\ 0 & f_{i} & v_{i} & 0\\ 0 & 0 & 0 & 1 \end{array}\right)\times \;R_{0}\times M_{x}$$

According to [4], $R_{0}$ is the rotation that accounts for the rectification of points in the reference camera and is the Euclidean transformation from LiDAR to the reference camera. When expressed in homogeneous coordinates, a projection matrix is a simple linear transformation that extends by multiplication from one vector space to another. The overview of our proposed methodology can be seen in the Figure 3.

### 3.3. Bird’s Eye View Representation

The height (H), intensity (I), and density (D) were used to encode the representation of a bird’s eye view. We must focus on a specific region of the point cloud to create a filter; we prefer to keep the points that pertain to our area of interest. We selected the directions that were slightly more compatible with the image axes because we viewed the data from the top, and we were interested in turning it into an image (see Algorithm 1). The projected point cloud was discretized into a 2D grid map with a 10 cm resolution. The maximum height of the points in the cell was used to calculate the height feature for each cell. The point cloud was evenly divided into slices to obtain more precise height information. The reflectance value of the point with the greatest height in each cell was the intensity feature. The density of a point cloud is the number of points in each cell. The density of all points mapped onto the grid map was normalized using the following formula:(3)$$D=\min\;\left(1.0,\;\frac{\log\left(k+1\right)}{\log\left(64\right)}\right),$$
where k is the number of points in a cell [10]. In summary, the bird’s eye view map is encoded as (n+2) channel features because the intensity and density features are computed for the entire point cloud, whereas the height feature is computed for n slices. We considered the point cloud area within the origin $A_{\mathrm{PC}}$ to be as follows:(4)APC={A=[x,y,z]^N }, | x∈(−20.0 m, 20.0 m), y ∈(0.0 m, 50.0 m), z ∈(−1.8 m, 1.5 m)
**Algorithm 1.** Pseudo-code for LiDAR to BEV conversion
start:  PointCloud2BEV function (points)input variables:  Resolution = 0.1  Side range = (−20, 20) # left side 20 m, right side 20 m view  Forward range = (0, 50) # we do not consider points that are behind the vehicle  Height range = (−1.8, 1.25) # assuming the position of LiDAR to be on 1.25 mprocessing:  Creates a 2D birds eye view representation of the point cloud data  Extract the points for each axis  Three filters for the front-to-back, side-to-side, and height ranges  Convert to pixel position values based on resolution  Shift pixels to have minimum be at the (0, 0)  Assign height values between a range (−1.8, 1.25 m)  Rescale the height values between 0 and 255  Fill the pixel values in an image arrayoutput:  Return BEV image

### 3.4. Bounding Box Derivation

We applied YOLOv4 [20], which is a modification of YOLOv3 [21] that includes the modified spatial attention module [22], path aggregation network [23], and spatial pyramid pooling block [24], which is divided into backbone (CSPDarknet53) [25], neck (SPP), and head blocks (YOLOv3) to provide better feature extraction. The boxes were predicted by the object detector for each BEV map. The angle regression increased the degrees of freedom or the number of potential priors, although we refrained from increasing the number of predictions because of efficiency concerns. Therefore, based on the distribution of anchor boxes in the KITTI dataset, we defined only three alternative sizes for vehicles, pedestrians, and cyclists. For the region-proposed network, we applied the Euler region proposal proposed (E-RPN) by Simon et al. [1]. The RPN was used to anticipate region proposals with a variety of scales and aspect ratios. The method avoids the need to provide images or filters with various sizes or aspect ratios by using a pyramid of regression. Based on the YOLOv4 regression parameters [26] and a complicated angle for the box orientation, we predicted oriented 2D bounding boxes and switched them to 3D, because we predetermined the height based on each detected class. To project 3D boxes onto an image, we must perform the following steps:

Limit the complex angle to be in boundaries −π/2≤θ ≤ π/2;Inverse the rigid body transformation matrix (3 × 4 as [R|t])([R′|−R′t; 0|1]);Recreate a 3D bounding box in the Velodyne coordinate system and then convert it to camera coordinates. The translational difference between the sensors is estimated by considering the following matrix multiplication:(5)$$M_{projective}=\;\left(\begin{array}{cccc} f & 0 & u_{0} & 0\\ 0 & f & v_{0} & 0\\ 0 & 0 & 0 & 1 \end{array}\right)\times \left(\begin{array}{cccc} 1 & 0 & 0 & t_{x}\\ 0 & 1 & 0 & t_{y}\\ 0 & 0 & 1 & t_{z}\\ 0 & 0 & 0 & 1 \end{array}\right)$$The target boxes are rescaled, yaw angles are obtained, and the box is drawn on the image.

### 3.5. Distance Estimation

For distance estimation, we applied triangulation distance measurements. The basis of the triangulation depends on the projection of light; more specifically, the camera records the laser projecting a line across an item [27]. Similarly, we assumed our car to be placed at the top middle point, and the projected point cloud data were the laser points in the image. Moreover, we did not validate the objects located behind the car. Figure 4 illustrates the field of view projected onto the BEV map image. The object at the top represents the car, and A, B, and C are the detected cars. The height of the BEV map with a resolution of 0.1 m per pixel is equivalent to *D* = 50 m, and with the width *W* = 40 m, as shown in Equation (3), we note that x∈(−20.0 m, 20.0 m). The pixel-wise distances to objects A, B, and C were defined as $d_{1}$, $d_{2}$, and $d_{3}$, respectively. The distance between the object and the system was determined using trigonometry.

### 3.6. Evaluation

Our objective was to obtain the estimated distance that is as close to the real distance as possible. The KITTI dataset does not contain labeled distance information for objects. Therefore, in this study, we manually estimated the distance from the visualized point cloud data. To obtain the closest (x, y, and z) points of the object in a scene, we used a visualizing tool, called SUSTecH Points [28]. We presumed our extracted points to be the “actual” distances of the object and our derived distances as “predicted” ones. We used five regression evaluation metrics to compare the performances of the proposed methodology. Regression evaluation methods are called error metrics. The absolute relative difference (6), squared relative difference (7), mean square errors (8), root of mean square errors (9), and root of mean squared computed from the log of the projected distance and log ground truth distance are the four metrics used to assess depth prediction (10).
(6)$$AbsRel=\;\frac{1}{n}\overset{n}{\underset{i=1}{\sum }}\frac{\left|p_{i}-a_{i}\right|}{a_{i}}$$
(7)$$SquaRel=\frac{1}{n}\;\overset{n}{\underset{i=1}{\sum }}\frac{{\left|\left|p_{i}-a_{i}\right|\right|}^{2}}{a_{i}}$$
(8)$$MSE=\;\frac{1}{n}\sum_{i=1}^{n}{\left(p_{i}\;-\;a_{i}\right)}^{2}$$
(9)$$RMSE=\sqrt{\frac{1}{n}\sum_{i=1}^{n}{\left(p_{i}\;-\;a_{i}\right)}^{2}}$$
(10)$$RMSLE=\;\sqrt{\frac{1}{n}\sum_{i=1}^{n}{\left(\log\left(p_{i}+1\right)-\log\left(a_{i}+1\right)\right)}^{2}}$$

Equations (6)–(10) are the evaluation techniques, also referred to as quantifiers of prediction error, where $a_{i}$ is the desired outcome, for example, i and $p_{i}$ is the predicted value from the model. We used a coverage area of 0–50 m, because we did not consider objects located farther than that distance.

## 4. Experiments and Results

In this section, we present our experimental setup with the results and comparisons with other suggested methods. For this experiment, we randomly selected 40 scenes containing 104 detected objects. The proposed Yolov4 detection model detects three classes: “car,” “pedestrian”, and “cyclist”. We selected scenes with distinct object positions and classified them into close-, middle-, and long-ranging objects, as shown in Figure 5. Therefore, the selected verification image should include object samples under various up-close and far-away conditions. Table 1 lists the quantitative results of the distance estimation method for the following four different regression evaluation metrics: AbsRel, SquaRel, RMSE, and RMSLE. The results are presented in terms of the following distance ranges: 0–10, 10–20, 20–30, 30–40, and 40–50 m. The overall results presented at the bottom suggest that the distance estimation method performs well, with low errors across all metrics and distance ranges. The errors are particularly low for distances between 20 and 30 m, with the lowest values for all metrics. We compared the proposed distance estimation system with other distance estimation methods from different perspectives, namely, error verification of the entire system model and system robustness verification, to confirm the accuracy and robustness of the system. Table 2 lists the results of the distance accuracy verification experiment for an autonomous vehicle. The experiment measured the distance estimation of the vehicle in various on-road situations and compared it with the actual distance. The table lists the actual distance, estimated distance, absolute error in meters, relative error in percentages, and average error in meters and percentages. Figure 5 shows the images of the associated scenes in Table 2. It is noteworthy that the relative error varies significantly depending on the scene, with some scenes having a relative error as low as 0.8% and others having a relative error as high as 5.88%. The experiment also showed that the relative error was less than 5%, which is considered an acceptable error threshold for most autonomous vehicle systems. Table 3 illustrates a comparison of our network’s quantitative performance with existing monocular, stereo, and LiDAR-based techniques on the KITTI dataset.

Table 3 contains the data of the head-to-head error metric and accuracy comparisons with the four different methodologies for the KITTI dataset. To validate the accuracy of our distance estimator, we used the threshold accuracy, which can be formulated as follows:(11)$$\%\;of\;\delta \;<\;1.25=>\;\delta =\;\max\left(\frac{p_{i}}{a_{i}},\;\frac{a_{i}}{p_{i}}\right)$$

Table 4 presents the threshold accuracy shown in Equation (11) for the various distance ranges. The proposed method proves that the proposed distance prediction framework is a highly accurate method for predicting distances on-road, outperforming existing methods across a wide range of distances. The framework is particularly accurate for mid-range distances, with an accuracy of 100% for distances between 10 and 50 m. However, because we did not consider objects farther than 50 m, we could not validate those distance regions.

## 5. Discussion

Figure 6 represents the result of our proposed distance estimation, wherein the blue line represents all actual 104 distances in ascending order, and the orange line represents the output distances from our triangulation method. The black lines connecting the points represent the absolute errors in meters projected on the graph. In contrast to the other methods, our method was more robust for long-distance range objects. Table 3 lists evidence that our method was more accurate than the previous methods. Our method was robust in terms of the detection of larger objects, such as cars. By contrast, our possible limitation is that our triangulation-based method is highly dependent on object detection. For example, Figure 7 shows the distance estimation for the “pedestrian” and “cyclist” classes. We can observe that the projected bounding box in (b) is slightly larger than the object, which can cause a higher distance estimation error. To address this limitation, we can apply better 3D object detectors in future work.

Although the number of detected objects was unknown in other research, we calculated the exact number of detections. As listed in Table 3, the proposed methodology achieved accuracies of 0.980, 1.000, and 1.000 in terms of δ<1.25, $\delta <{\left(1.25\right)}^{2}$ and $\delta <{\left(1.25\right)}^{3}$, respectively, indicating that it was among the finest in terms of accuracy and robustness. Our proposed methodology had a significantly lower error rate for known metrics and a higher accuracy rate, except for the accuracy when δ<1.25. The method proposed by Lee et al. performed slightly better, because our method had a larger error rate at distances ranging from 10 to 20 m, with 91.6% for close-range cars. Table 4 lists a head-to-head accuracy comparison with previous studies. Our work was slightly less accurate in estimating the distance to the close objects, because we had an average absolute error of 0.4 m, and closer objects had a higher error rate. Figure 6 shows that objects located in a range from approximately 6 to 12 m had greater error rates, because we can distinguish between “actual” and “predicted” points.

## 6. Conclusions and Future Work

In this study, we proposed a methodology that uses point cloud data captured by LiDAR positioned in the car and converts it into a BEV map to estimate the distances between a vehicle and objects in front of the vehicle. The proposed method consists of a 3D object detector to derive the possible bounding boxes for the detected classes, and then projects 3D boxes onto a 2D BEV map using the angular regression algorithm. After the bounding boxes were projected onto the image, we applied the triangulation method, which is widely used as a distance estimator in several applications. Consequently, our novel method proved to be efficient for on-road comparisons with other existing methodologies, achieving an overall accuracy of 98.0%. In addition, our method proved to be 100% accurate in the distance range of 10–50 m. As mentioned previously, our approach was efficient in identifying larger objects and relied heavily on the power of the object detector. We demonstrated that the bounding boxes for smaller object classes can be slightly larger than the object, leading to a higher error rate in the distance estimation. With a 2.3% error rate in the distance estimation, our methodology may be acceptable in cases wherein the vehicle is driving on a highway with relatively few obstacles. However, the acceptable error level depends on the safety and performance requirements of a specific application. In a future work, we will apply a more accurate 3D object detector to address these limitations. The practical application of our methodology can be used in self-driving cars, drones, and robotics, wherein accurate distance estimation is essential for safe navigation and decision making. The proposed method can also be used in other applications, such as object detection, localization, and tracking in different domains, such as security, surveillance, and augmented reality.

## Figures and Tables

**Figure 1 sensors-23-02103-f001:**
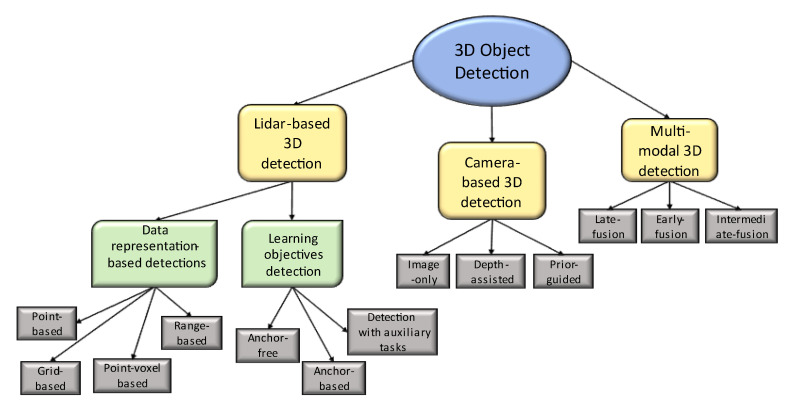
3D object detection hierarchy.

**Figure 2 sensors-23-02103-f002:**
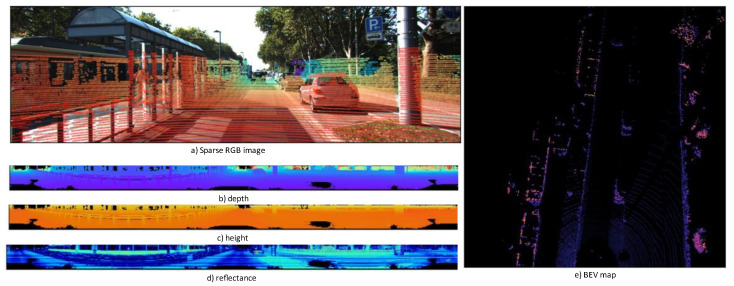
Multiview representation of a 3D point cloud projected into RGB images.

**Figure 3 sensors-23-02103-f003:**
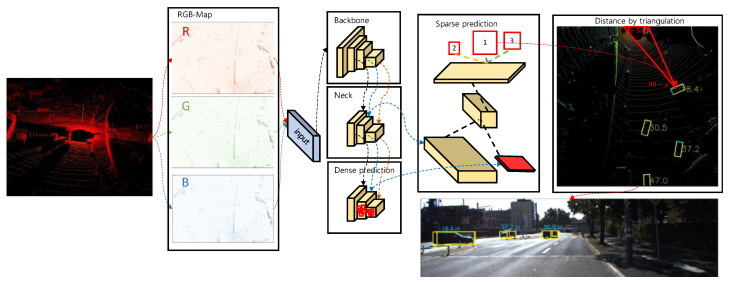
Overview of the distance extraction from the LiDAR data. First, point cloud data are classified according to height, intensity, and density. Then, this data is fed into the convolutional network (YOLOv4) to obtain bounding 2D boxes on the BEV map. Subsequently, as described in Section 3.4, 2D boxes are projected as 3D boxes onto the image (1,2 and 3 are the detected objects on the image).

**Figure 4 sensors-23-02103-f004:**
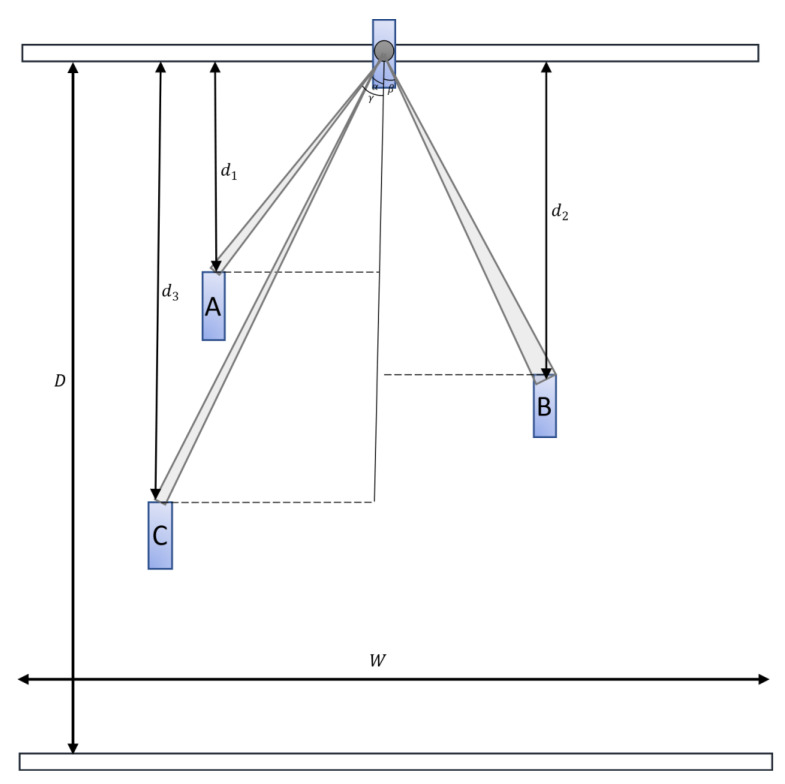
General overview of the distance estimation. A, B, C are the cars located in distances *d*_1_, *d*_2_, and *d*_3_ respectively from the origin by y-axis.

**Figure 5 sensors-23-02103-f005:**
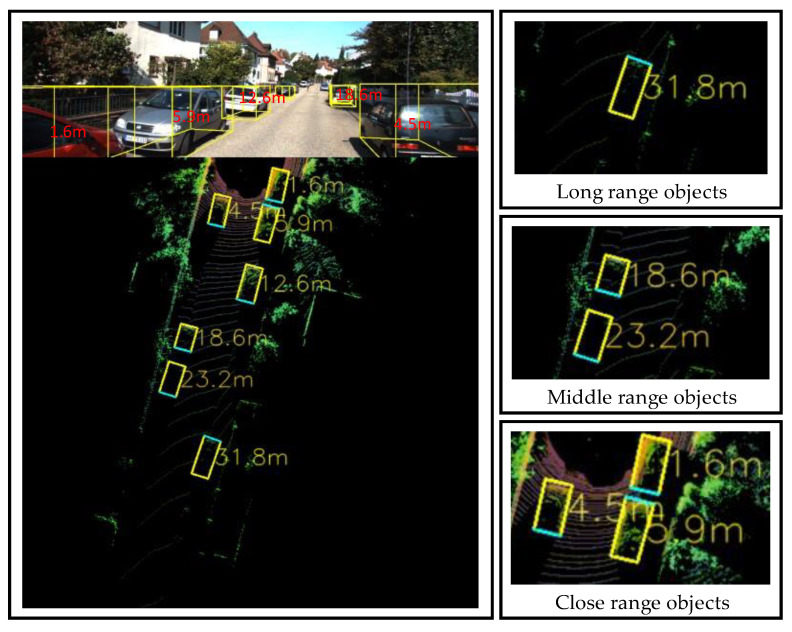
Distance estimation results (in meters) displayed onto the BEV map.

**Figure 6 sensors-23-02103-f006:**
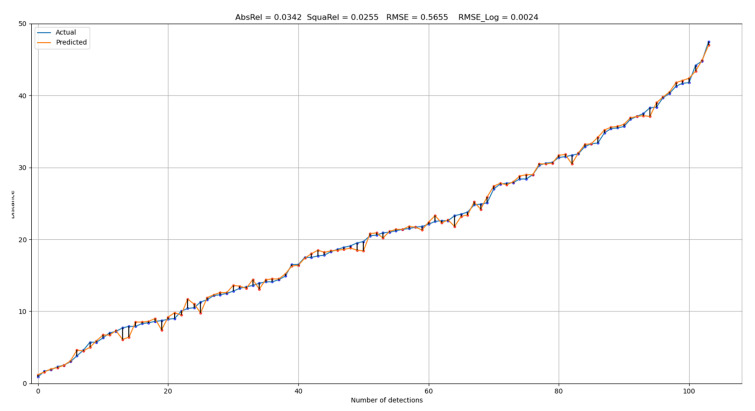
Distance estimation results of the difference between “actual” and “predicted” for all detections.

**Figure 7 sensors-23-02103-f007:**
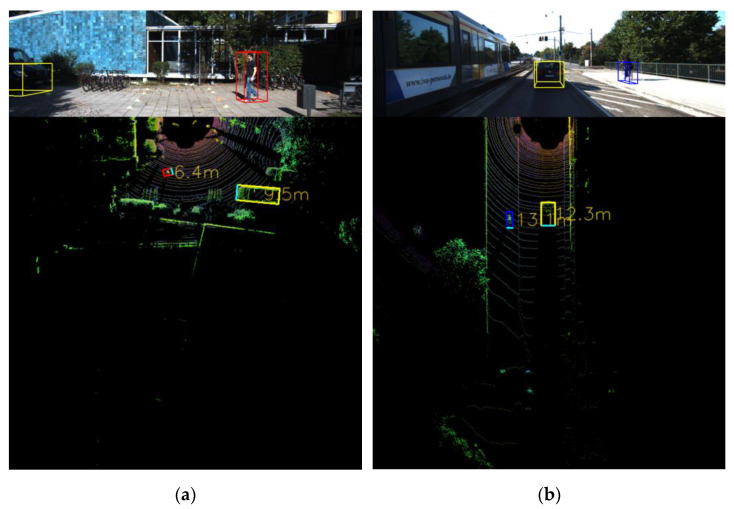
Distance estimation for different class objects: (**a**) car and pedestrian; (**b**) car and cyclist.

**Table 1 sensors-23-02103-t001:** Quantitative results of our distance estimation method for the four regression evaluation metrics in terms of various distances ranges.

Distance (m)	AbsRel	SquaRel	RMSE	RMSLE
0–10	0.0836	0.0615	0.6477	0.0078
10–20	0.0337	0.0282	0.6208	0.0019
20–30	0.0160	0.0105	0.4969	0.0004
30–40	0.0100	0.0068	0.4843	0.0002
40–50	0.0103	0.0056	0.4843	0.0002
**Overall**	0.0342	0.0255	0.5655	0.0024

**Table 2 sensors-23-02103-t002:** Distance accuracy verification by comparing ground truth data with the estimated distances.

	Number of Detections per Scene	Car 1	Car 2	Car 3	Car 4	Car 5	Car 6	Car 7
**Distance Verification**	Actual (m)	1.7	4.6	5.7	12.5	18.9	23.5	31.5
	Estimated (m)	1.6	4.5	5.9	12.6	18.6	23.2	31.8
	Absolute error (m)	0.1	0.1	0.2	0.1	0.3	0.3	0.3
	Relative error (%)	5.88	2.17	3.51	0.80	1.59	1.28	0.95
	**Average error (m)**	**0.2**						
	**Average error (%)**	**2.31**						

**Table 3 sensors-23-02103-t003:** Error metric and accuracy comparison of the quantitative performance of the proposed methodology with those of the existing techniques on the KITTI dataset.

Study	Methodology			Error Metric				Accuracy (in %)		
	Stereo	Monocular	LiDAR	*AbsRel*	*SquaRel*	RMSE	RMSLE	δ<1.25	$\delta <1.25^{2}$	$\delta <1.25^{3}$
Lee et al. [18]		✓		0.047	0.116	2.091	0.076	0.982	0.996	1.000
Zhou et al. [29]		✓		0.183	×	6.709	0.270	0.734	0.902	0.959
Ding et al. [30]	✓			0.071	×	3.740	×	0.934	0.979	0.992
Liang et al. [31]		✓		0.101	0.715	×	0.178	0.899	0.981	0.990
**Ours**			✓	**0.0342**	**0.0255**	**0.5655**	**0.0024**	**0.980**	**1.000**	**1.000**

**Table 4 sensors-23-02103-t004:** Threshold accuracy results (in % δ<1.25 ) of the proposed distance prediction framework’s on-road evaluation at various distance ranges.

Distance (m)	Ours	Lee et al. [18]	Kim et al. [32]	Kumar et al. [19]
0–10	0.913	**0.983**	0.980	×
10–20	**1.000**	0.987	0.922	×
20–30	**1.000**	0.984	0.917	0.980
30–40	**1.000**	0.995	0.913	×
40–50	**1.000**	0.975	0.912	0.963
50–60	×	0.974	×	×
60–70	×	0.931	×	×
70–80	×	0.963	×	0.960

## Data Availability

The data used in this paper can be found in https://www.cvlibs.net/datasets/kitti/ (accessed on 4 April 2012).

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
