# Peer review of "Accurate 3D to 2D Object Distance Estimation from the Mapped Point Cloud Data"

_sensors, 2023, doi:10.3390/s23042103_

Round 1

Reviewer 1 Report

Comments to Author

The author proposed a methodology a reliable map-based 11 bird’s eye view (BEV) that calculates the distance to the detected objects. With the help of the E-RPN 12 model, a LiDAR-to-image-based method for metric distance estimation with 3D bounding box projections onto the image. The paper is good, but author fails to present the system design and the discussion of results. My comments to enhance the paper as follows:

1) Title should be corrected "3Dto2D" to be "3D to 2D", "2 point" to "2-point"

2) Abstract should clearly mention the challenge of the 3D to 2D object distance estimation from the mapped 2-point cloud data.

3) The challenges/problem statement should be explained at the end of section 1.

4) Change the title of section 3 into Proposed or System Model

5) Add structure diagram- or flowchart diagram for your proposed model. Also, enhance the grammar of algorithm 1. For example, "Fix height values to between".

6) In the Discussion, you have to justify why your method is better than the baseline methods.

7) Is it logic that your method achieve 100% accurate in distance ranges from 10 to 50 meters?

Reviewer 2 Report

This paper proposed “Accurate 3Dto2D object distance estimation from the mapped 2-point cloud data.”. The approach discussed in this manuscript is interesting. I recommend following corrections.

1-     Related work needs more investigation of some latest and relevant work

2-     The introduction section needs more investigation of some recent and relevant work that has been done in the past. I suggest a few papers for your reference.  (Lidar Point Cloud Compression, Processing and Learning for Autonomous Driving, ITITS-2022, A Step toward Next-Generation Advancements in the Internet of Things Technologies”)

3-     Briefly described your proposed methodology and dataset.

4-     Clearly defined your manuscript motivation and practical application

5-     Add Pseudocode with Examples of your proposed and based method. What is your contribution.

6-      The experiment section provides additional information, justifies your work, and compares it to other states or arts methods. 

7-     Most of the references in the Bibliography sections are quite old. Recent references may be used.

8-     In the manuscript, there are many grammatical errors and typos. Carefully revised all manuscripts and corrected them.

9-     Although it is not required, we encourage you to include your source code with your manuscript to demonstrate the origin of your work.

Reviewer 3 Report

The authors researched a novel distance estimation method that works with converted point cloud data. A so-called KITTI dataset is used (it is also used in autonomous car research) because it contains LIDAR ground truth data. The LiDAR map is transformed into a bird's eye view. To estimate the distance, the triangulation method is used, which is based on knowing the known image width (in meters) and image height (in meters), and the distance to the object is calculated through trigonometric transformations. It should be noted that the width of the LiDAR map is limited to 40m and the height to 50m. They ignore other data. The YOLO network only serves to obtain a 2D bounding box on the image, which is converted into a 3D bounding box. Using the transformed LiDAR map, the distance to the object is calculated, and the result is mapped onto the 3D bounding box in the image.

This is interesting research. In my opinion, the manuscript is well-structured, easy to follow, and well-written.

Before the publication, I have a few minor comments for the authors to address it.

  1. Please define the abbreviation E-RPN (L12 and L89).
  2. L255, Please, add the abbreviation for the region-proposed network.
  3. Please, add a sentence about future research in the conclusion.

Round 2

Reviewer 2 Report

No more comments.